# Image Augmentation Is All You Need: Regularizing Deep Reinforcement Learning from Pixels

**Denis Yarats**[*]
New York University & Facebook AI Research
denisyarats@cs.nyu.edu

**Ilya Kostrikov**[*]
New York University
kostrikov@cs.nyu.edu

**Rob Fergus**
New York University
fergus@cs.nyu.edu

## Abstract

Existing model-free reinforcement learning (RL) approaches are effective when trained on states but struggle to learn directly from image observations. We propose an augmentation technique that can be applied to standard model-free RL algorithms, enabling robust learning directly from pixels without the need for auxiliary losses or pre-training. The approach leverages input perturbations commonly used in computer vision tasks to transform input examples, as well as regularizing the value function and policy. Our approach reaches a new state-of-the-art performance on DeepMind control suite and Atari 100k benchmark, surpassing previous model-free (Haarnoja et al., 2018; van Hasselt et al., 2019a), model-based (Hafner et al., 2019; Lee et al., 2019; Hafner et al., 2018; Kaiser et al., 2019) and contrastive learning (Srinivas et al., 2020) approaches. It also closes the gap between state-based and image-based RL training. Our method, which we dub **DrQ**: **D**ata-**r**egularized **Q**, can be combined with any model-free RL algorithm. To the best of our knowledge, our approach is the first effective data augmentation method for RL on these benchmarks.

## 1 Introduction

Sample-efficient deep reinforcement learning (RL) algorithms capable of directly training from image pixels would open up many real-world applications in control and robotics. However, simultaneously training a convolutional encoder alongside a policy network is challenging when given limited environment interaction, strong correlation between samples and a typically sparse reward signal. Limited supervision is a common problem across AI and two approaches are commonly taken: (i) training with an additional auxiliary losses, such as those based on self-supervised learning (SSL) and (ii) training with data augmentation.

A wide range of auxiliary loss functions have been proposed to augment supervised objectives, e.g. weight regularization, noise injection (Hinton et al., 2012), or various forms of auto-encoder (Kingma et al., 2014). In RL, reconstruction losses (Jaderberg et al., 2017; Yarats et al., 2019) or SSL objectives (Dwibedi et al., 2018; Srinivas et al., 2020) are used. However, these objectives are unrelated to the task at hand, thus have no guarantee of inducing an appropriate representation for the policy network. SSL losses are highly effective in the large data regime, e.g. in domains such as vision (Chen et al., 2020; He et al., 2019) and NLP (Collobert et al., 2011; Devlin et al., 2018) where large (unlabeled) datasets are readily available. However, in sample-efficient RL, training data is more limited due to restricted interaction between the agent and the environment, limiting their effectiveness.

Data augmentation methods are widely used in vision and speech domains, where output-invariant perturbations can easily be applied to the labeled input examples. Surprisingly, data augmentation

---

[*]Equal contribution. Author ordering determined by coin flip. Both authors are corresponding.

has received little attention in the RL community. In this paper we propose augmentation approaches appropriate for sample-efficient RL and comprehensively evaluate them. The key idea of our approach is to use standard image transformations to perturb input observations, as well as regularizing the $Q$-function learned by the critic so that different transformations of the same input image have similar $Q$-function values. No further modifications to standard actor-critic algorithms are required. Our study is, to the best of our knowledge, the first careful examination of image augmentation in sample-efficient RL.

The main contributions of the paper are as follows: (i) the first to demonstrate that data augmentation greatly improves performance when training model-free RL algorithms from images; (ii) introducing a natural way to exploit MDP structure through two mechanisms for regularizing the value function, in a manner that is generally applicable to model-free RL and (iii) setting a new state-of-the-art performance on the standard DeepMind control suite (Tassa et al., 2018), closing the gap between learning from states, and Atari 100k (Kaiser et al., 2019) benchmarks.

## 2 RELATED WORK

**Data Augmentation in Computer Vision** Data augmentation via image transformations has been used to improve generalization since the inception of convolutional networks (Becker & Hinton, 1992; Simard et al., 2003; LeCun et al., 1989; Ciresan et al., 2011; Ciregan et al., 2012). Following AlexNet (Krizhevsky et al., 2012), they have become a standard part of training pipelines. For object classification tasks, the transformations are selected to avoid changing the semantic category, i.e. translations, scales, color shifts, etc. While a similar set of transformations are potentially applicable to control tasks, the RL context does require modifications to be made to the underlying algorithm.

Data augmentation methods have also been used in the context of self-supervised learning. Dosovitskiy et al. (2016) use per-exemplar perturbations in a unsupervised classification framework. More recently, several approaches (Chen et al., 2020; He et al., 2019; Misra & van der Maaten, 2019) have used invariance to imposed image transformations in contrastive learning schemes, producing state-of-the-art results on downstream recognition tasks. By contrast, our scheme addresses control tasks, utilizing different types of invariance.

**Data Augmentation in RL** In contrast to computer vision, data augmentation is rarely used in RL. Certain approaches implicitly adopt it, for example Levine et al. (2018); Kalashnikov et al. (2018) use image augmentation as part of the AlexNet training pipeline without analysing the benefits occurring from it, thus being overlooked in subsequent work. HER (Andrychowicz et al., 2017) exploits information about the observation space by goal and reward relabeling, which can be viewed as a way to perform data augmentation. Other work uses data augmentation to improve generalization in domain transfer (Cobbe et al., 2018). However, the classical image transformations used in vision have not previously been shown to definitively help on standard RL benchmarks. Concurrent with our work, RAD (Laskin et al., 2020) performs an exploration of different data augmentation approaches, but is limited to transformations of the image alone, without the additional augmentation of the Q-function used in our approach. Moreover, RAD can be regarded as a special case of our algorithm. Multiple follow ups to our initial preprint appeared on ArXiv (Raileanu et al., 2020; Okada & Taniguchi, 2020), using similar techniques on other tasks, thus supporting the effectiveness and generality of data augmentation in RL.

**Continuous Control from Pixels** There are a variety of methods addressing the sample-efficiency of RL algorithms that directly learn from pixels. The most prominent approaches for this can be classified into two groups, model-based and model-free methods. The model-based methods attempt to learn the system dynamics in order to acquire a compact latent representation of high-dimensional observations to later perform policy search (Hafner et al., 2018; Lee et al., 2019; Hafner et al., 2019). In contrast, the model-free methods either learn the latent representation indirectly by optimizing the RL objective (Barth-Maron et al., 2018; Abdolmaleki et al., 2018) or by employing auxiliary losses that provide additional supervision (Yarats et al., 2019; Srinivas et al., 2020; Sermanet et al., 2018; Dwibedi et al., 2018). Our approach is complementary to these methods and can be combined with them to improve performance.

## 3 BACKGROUND

**Reinforcement Learning from Images** We formulate image-based control as an infinite-horizon partially observable Markov decision process (POMDP) (Bellman, 1957; Kaelbling et al., 1998). An POMDP can be described as the tuple $(\mathcal{O}, \mathcal{A}, p, r, \gamma)$, where $\mathcal{O}$ is the high-dimensional observation space (image pixels), $\mathcal{A}$ is the action space, the transition dynamics $p = Pr(o'_t|o_{\leq t}, a_t)$ capture the probability distribution over the next observation $o'_t$ given the history of previous observations $o_{\leq t}$ and current action $a_t$, $r : \mathcal{O} \times \mathcal{A} \to \mathbb{R}$ is the reward function that maps the current observation and action to a reward $r_t = r(o_{\leq t}, a_t)$, and $\gamma \in [0, 1)$ is a discount factor. Per common practice (Mnih et al., 2013), throughout the paper the POMDP is converted into an MDP (Bellman, 1957) by stacking several consecutive image observations into a state $s_t = \{o_t, o_{t-1}, o_{t-2}, \ldots\}$. For simplicity we redefine the transition dynamics $p = Pr(s'_t|s_t, a_t)$ and the reward function $r_t = r(s_t, a_t)$. We then aim to find a policy $\pi(a_t|s_t)$ that maximizes the cumulative discounted return $\mathbb{E}_\pi[\sum_{t=1}^{\infty} \gamma^t r_t | a_t \sim \pi(\cdot|s_t), s'_t \sim p(\cdot|s_t, a_t), s_1 \sim p(\cdot)]$.

**Soft Actor-Critic** The Soft Actor-Critic (SAC) (Haarnoja et al., 2018) learns a state-action value function $Q_\theta$, a stochastic policy $\pi_\theta$ and a temperature $\alpha$ to find an optimal policy for an MDP $(\mathcal{S}, \mathcal{A}, p, r, \gamma)$ by optimizing a $\gamma$-discounted maximum-entropy objective (Ziebart et al., 2008). $\theta$ is used generically to denote the parameters updated through training in each part of the model.

**Deep Q-learning** DQN (Mnih et al., 2013) also learns a convolutional neural net to approximate Q-function over states and actions. The main difference is that DQN operates on discrete actions spaces, thus the policy can be directly inferred from Q-values. In practice, the standard version of DQN is frequently combined with a set of refinements that improve performance and training stability, commonly known as Rainbow (van Hasselt et al., 2015). For simplicity, the rest of the paper describes a generic actor-critic algorithm rather than DQN or SAC in particular. Further background on DQN and SAC can be found in Appendix A.

## 4 SAMPLE EFFICIENT REINFORCEMENT LEARNING FROM PIXELS

### 4.1 OPTIMALITY INVARIANT IMAGE TRANSFORMATIONS FOR Q FUNCTION

We first introduce a general framework for regularizing the value function through transformations of the input state. For a given task, we define an optimality invariant state transformation $f : \mathcal{S} \times \mathcal{T} \to \mathcal{S}$ as a mapping that preserves the $Q$-values

$$Q(s, a) = Q(f(s, \nu), a) \text{ for all } s \in \mathcal{S}, a \in \mathcal{A} \text{ and } \nu \in \mathcal{T}.$$

where $\nu$ are the parameters of $f(\cdot)$, drawn from the set of all possible parameters $\mathcal{T}$. One example of such transformations are the random image translations successfully applied in the previous section.

For every state, the transformations allow the generation of several surrogate states with the same $Q$-values, thus providing a mechanism to reduce the variance of $Q$-function estimation. In particular, for an arbitrary distribution of states $\mu(\cdot)$ and policy $\pi$, instead of using a single sample $s^* \sim \mu(\cdot)$, $a^* \sim \pi(\cdot|s^*)$ estimation of the following expectation

$$\mathbb{E}_{\substack{s \sim \mu(\cdot) \\ a \sim \pi(\cdot|s)}} [Q(s, a)] \approx Q(s^*, a^*)$$

we generate $K$ samples via random transformations and obtain an estimate with lower variance

$$\mathbb{E}_{\substack{s \sim \mu(\cdot) \\ a \sim \pi(\cdot|s)}} [Q(s, a)] \approx \frac{1}{K} \sum_{k=1}^{K} Q(f(s^*, \nu_k), a_k) \text{ where } \nu_k \in \mathcal{T} \text{ and } a_k \sim \pi(\cdot|f(s^*, \nu_k)).$$

This suggests two distinct ways to regularize $Q$-function. First, we use the data augmentation to compute the target values for every transition tuple $(s_i, a_i, r_i, s'_i)$ as

$$y_i = r_i + \gamma \frac{1}{K} \sum_{k=1}^{K} Q_\theta(f(s'_i, \nu'_{i,k}), a'_{i,k}) \text{ where } a'_{i,k} \sim \pi(\cdot|f(s'_i, \nu'_{i,k})) \tag{1}$$

where $\nu'_{i,k} \in \mathcal{T}$ corresponds to a transformation parameter of $s'_i$. Then the Q-function is updated using these targets through an SGD update using learning rate $\lambda_\theta$

$$\theta \leftarrow \theta - \lambda_\theta \nabla_\theta \frac{1}{N} \sum_{i=1}^{N} (Q_\theta(f(s_i, \nu_i), a_i) - y_i)^2. \tag{2}$$

In tandem, we note that the same target from Equation (1) can be used for different augmentations of $s_i$, resulting in the second regularization approach

$$\theta \leftarrow \theta - \lambda_\theta \nabla_\theta \frac{1}{NM} \sum_{i=1,m=1}^{N,M} (Q_\theta(f(s_i, \nu_{i,m}), a_i) - y_i)^2. \tag{3}$$

When both regularization methods are used, $\nu_{i,m}$ and $\nu'_{i,k}$ are drawn independently.

### 4.2 PRACTICAL INSTANTIATION OF OPTIMALITY INVARIANT IMAGE TRANSFORMATION

A range of successful image augmentation techniques have been developed in computer vision (Ciregan et al., 2012; Ciresan et al., 2011; Simard et al., 2003; Krizhevsky et al., 2012; Chen et al., 2020). These apply transformations to the input image for which the task labels are invariant, e.g. for object recognition tasks, image flips and rotations do not alter the semantic label. However, tasks in RL differ significantly from those in vision and in many cases the reward would not be preserved by these transformations. We examine image transformations from Chen et al. (2020) (random shifts, random cutouts, horizontal/vertical flips, rotations and intensity shifts) in Appendix E and conclude that random shifts strike a good balance between simplicity and performance, we therefore limit our choice of transformation function $f(\cdot)$ to random shifts.

We apply shifts to the images sampled from the replay buffer. For example, images from the DeepMind control suite used in our experiments are $84 \times 84$. We pad each side by $4$ pixels (by repeating boundary pixels) and then select a random $84 \times 84$ crop, yielding the original image shifted by $\pm 4$ pixels. This procedure is repeated every time an image is sampled from the replay buffer.

### 4.3 OUR APPROACH: DATA-REGULARIZED Q (DRQ)

Our approach, **DrQ**, is the union of the three separate regularization mechanisms introduced above:

1. transformations of the input image (Section 4.2).
2. averaging the $Q$ target over K image transformations (Equation (1)).
3. averaging the $Q$ function itself over M image transformations (Equation (3)).

Algorithm 1 details how they are incorporated into a generic pixel-based off-policy actor-critic algorithm. Note that if [K=1,M=1] then **DrQ** reverts to *image transformations* alone, this makes applying **DrQ** to any model-free RL algorithm straightforward.

For the experiments in this paper, we pair **DrQ** with SAC (Haarnoja et al., 2018) and DQN (Mnih et al., 2013), popular model-free algorithms for control in continuous and discrete action spaces respectively. We select image shifts as the class of image transformations $f$, with $\nu \pm 4$, as explained in Section 4.2.

## 5 EXPERIMENTS

### 5.1 ABLATION EXPERIMENT

Figure 1 shows the effect of image shift augmentation applied to three tasks from the DeepMind control suite (Tassa et al., 2018). Figure 1a shows unmodified SAC (Haarnoja et al., 2018) parameterized with different image encoders, taken from: NatureDQN (Mnih et al., 2013), Dreamer (Hafner et al., 2019), Impala (Espeholt et al., 2018), SAC-AE (Yarats et al., 2019), and D4PG (Barth-Maron et al., 2018). The encoders vary significantly in their architecture and capacity, with parameter

---

**Algorithm 1 DrQ**: **D**ata-**r**egularized **Q** applied to a generic off-policy actor critic algorithm.
Black: unmodified off-policy actor-critic.
Orange: image transformation.
Green: target $Q$ augmentation.
Blue: $Q$ augmentation.

---

**Hyperparameters:** Total number of environment steps $T$, mini-batch size $N$, learning rate $\lambda_\theta$, target network update rate $\tau$, image transformation $f$, number of target $Q$ augmentations $K$, number of $Q$ augmentations $M$.

**for** each timestep $t = 1..T$ **do**
    $a_t \sim \pi(\cdot|s_t)$
    $s'_t \sim p(\cdot|s_t, a_t)$
    $\mathcal{D} \leftarrow \mathcal{D} \cup (s_t, a_t, r(s_t, a_t), s'_t)$           $\triangleright$ Add a transition to the replay buffer
    UPDATECRITIC($\mathcal{D}$)
    UPDATEACTOR($\mathcal{D}$)   $\triangleright$ Data augmentation is applied to the samples for actor training as well
**end for**
**procedure** UPDATECRITIC($\mathcal{D}$)
    $\{(s_i, a_i, r_i, s'_i)\}_{i=1}^N \sim \mathcal{D}$          $\triangleright$ Sample a mini batch from the replay buffer
    $\left\{ \nu'_{i,k} \middle| \nu'_{i,k} \sim \mathcal{U}(\mathcal{T}), i = 1..N, k = 1..K \right\}$     $\triangleright$ Uniformly sample target augmentations
    **for** each $i = 1..N$ **do**
        $a'_i \sim \pi(\cdot|s'_i)$ or $a'_{i,k} \sim \pi(\cdot | f(s'_i, \nu'_{i,k})), k = 1..K$
        $\hat{Q}_i = Q_{\theta'}(s'_i, a'_i)$ or $\hat{Q}_i = \frac{1}{K} \sum_{k=1}^K Q_{\theta'}(f(s'_i, \nu'_{i,k}), a'_{i,k})$
        $y_i \leftarrow r(s_i, a_i) + \gamma \hat{Q}_i$
    **end for**
    $\{\nu_{i,m} | \nu_{i,m} \sim \mathcal{U}(\mathcal{T}), i = 1..N, m = 1..M\}$     $\triangleright$ Uniformly sample Q augmentations
    $J_Q(\theta) = \frac{1}{N} \sum_{i=1}^N (Q_\theta(s_i, a_i) - y_i)^2$ or $J_Q(\theta) = \frac{1}{NM} \sum_{i,m=1}^{N,M} (Q_\theta(f(s_i, \nu_{i,m}), a_i) - y_i)^2$
    $\theta \leftarrow \theta - \lambda_\theta \nabla_\theta J_Q(\theta)$            $\triangleright$ Update the critic
    $\theta' \leftarrow (1 - \tau)\theta' + \tau\theta$            $\triangleright$ Update the critic target
**end procedure**

---

counts ranging from 220k to 2.4M. None of these train satisfactorily, with performance decreasing for the larger capacity models. Figure 1b shows SAC with the application of our random shifts transformation of the input images (i.e. just Section 4.2, not Q augmentation also). The results for all encoder architectures improve dramatically, suggesting that our method is general and can assist many different encoder architectures. To the best of our knowledge, this is the first successful demonstration of applying image augmentation on the standard benchmarks for continuous control. Furthermore, Figure 2 shows the full **DrQ**, with both image shifts and Q augmentation (Section 4.1), as well as ablated versions. Q augmentation provides additional consistent gain over image shift augmentation alone (full results are in Appendix F).

## 5.2 DEEPMIND CONTROL SUITE EXPERIMENTS

In this section we evaluate our algorithm (**DrQ**) on the two commonly used benchmarks based on the DeepMind control suite (Tassa et al., 2018), namely the PlaNet (Hafner et al., 2018) and Dreamer (Hafner et al., 2019) setups. Throughout these experiments all hyper-parameters of the algorithm are kept fixed: the actor and critic neural networks are trained using the Adam optimizer (Kingma & Ba, 2014) with default parameters and a mini-batch size of $512$ [1]. For SAC, the soft target update rate $\tau$ is 0.01, initial temperature is 0.1, and target network and the actor updates are made every 2 critic updates (as in Yarats et al. (2019)). We use the image encoder architecture from SAC-AE (Yarats et al., 2019) and follow their training procedure. The full set of parameters can be found in Appendix B. Following Henderson et al. (2018), the models are trained using 10 different seeds; for every seed the mean episode returns are computed every 10000 environment steps, averaging over 10 episodes. All figures plot the mean performance over the 10 seeds, together with $\pm$ 1 standard deviation shading. We compare our **DrQ** approach to leading model-free and model-based

---

[1] Note that **DrQ** does not utilize additional information beyond transitions sampled from the replay buffer (i.e. does not use more observations per mini-batch), thus the mini-batch size is the same as for unmodified SAC.

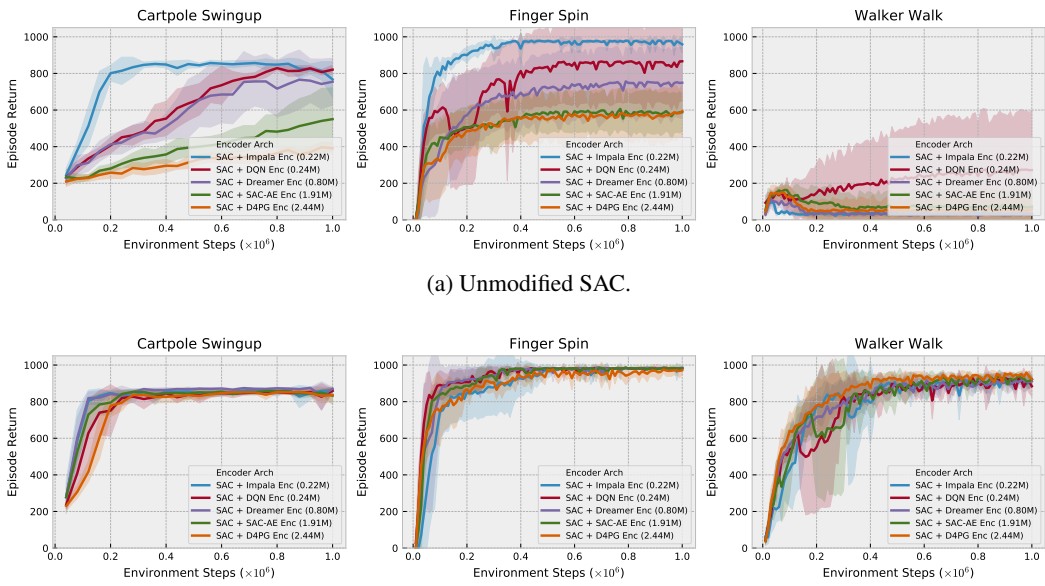

(a) Unmodified SAC.

(b) SAC with random shifts augmentation.

Figure 1: The performance of SAC trained from pixels on the DeepMind control suite using image encoder networks of different capacity (network architectures taken from recent RL algorithms, with parameter count indicated). **(a)**: unmodified SAC. Task performance can be seen to get worse as the capacity of the encoder increases. For Walker Walk (right), all architectures provide mediocre performance, demonstrating the inability of SAC to train directly from pixels on harder problems. **(b)**: SAC combined with image augmentation in the form of random shifts. The task performance is now similar for all architectures, regardless of their capacity, which suggests the generality of our method. There is also a clear performance improvement relative to (a), particularly for the more challenging Walker Walk task.

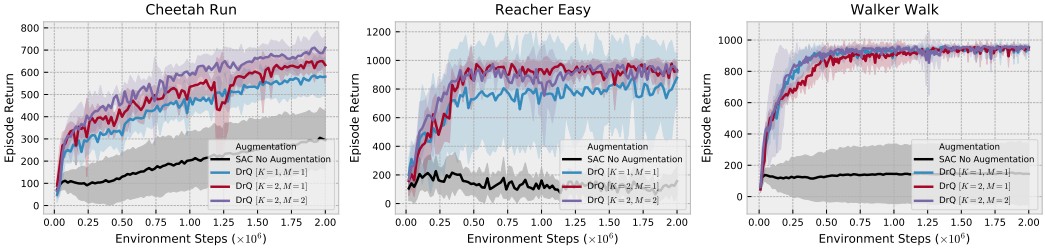

Figure 2: Different combinations of our three regularization techniques on tasks from (Tassa et al., 2018) using SAC. Black: standard SAC. Blue: **DrQ** [K=1,M=1], SAC augmented with random shifts. Red: **DrQ** [K=2,M=1], random shifts + Target Q augmentations. Purple: **DrQ** [K=2,M=2], random shifts + Target Q + Q augmentations. All three regularization methods correspond to Algorithm 1 with different K,M showing clear gains when both Target Q and Q augmentations are used.

approaches: PlaNet (Hafner et al., 2018), SAC-AE (Yarats et al., 2019), SLAC (Lee et al., 2019), CURL (Srinivas et al., 2020) and Dreamer (Hafner et al., 2019). The comparisons use the results provided by the authors of the corresponding papers.

**PlaNet Benchmark** (Hafner et al., 2018) consists of six challenging control tasks from (Tassa et al., 2018) with different traits. The benchmark specifies a different action-repeat hyper-parameter for each of the six tasks[2]. Following common practice (Hafner et al., 2018; Lee et al., 2019; Yarats et al.,

---

[2]This means the number of training observations is a fraction of the environment steps (e.g. an episode of 1000 steps with action-repeat 4 results in 250 training observations).

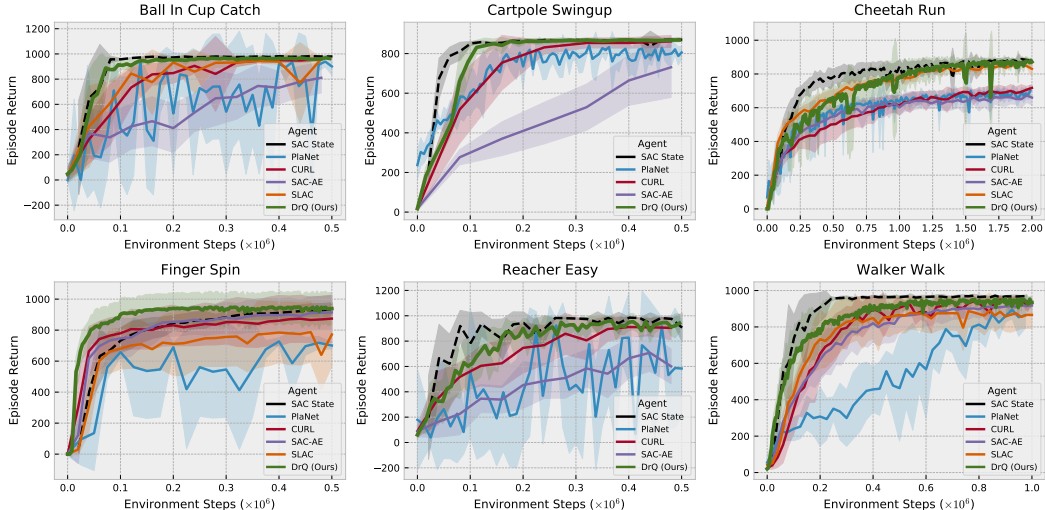

Figure 3: The PlaNet benchmark. Our algorithm (**DrQ** [K=2,M=2]) outperforms the other methods and demonstrates the state-of-the-art performance. Furthermore, on several tasks **DrQ** is able to match the upper-bound performance of SAC trained directly on internal state, rather than images. Finally, our algorithm not only shows improved sample-efficiency relative to other approaches, but is also faster in terms of wall clock time.

2019; Mnih et al., 2013), we report the performance using true environment steps, thus are invariant to the action-repeat hyper-parameter. Aside from action-repeat, all other hyper-parameters of our algorithm are fixed across the six tasks, using the values previously detailed.

Figure 3 compares **DrQ** [K=2,M=2] to PlaNet (Hafner et al., 2018), SAC-AE (Yarats et al., 2019), CURL (Srinivas et al., 2020), SLAC (Lee et al., 2019), and an upper bound performance provided by SAC (Haarnoja et al., 2018) that directly learns from internal states. We use the version of SLAC that performs one gradient update per an environment step to ensure a fair comparison to other approaches. **DrQ** achieves state-of-the-art performance on this benchmark on all the tasks, despite being much simpler than other methods. Furthermore, since **DrQ** does not learn a model (Hafner et al., 2018; Lee et al., 2019) or any auxiliary tasks (Srinivas et al., 2020), the wall clock time also compares favorably to the other methods.

In Table 1 we also compare performance given at a fixed number of environment interactions (e.g. 100k and 500k). Furthermore, in Appendix G we demonstrate that **DrQ** is robust to significant changes in hyper-parameter settings.

**Dreamer Benchmark** is a more extensive testbed that was introduced in Dreamer (Hafner et al., 2019), featuring a diverse set of tasks from the DeepMind control suite. Tasks involving sparse reward were excluded (e.g. Acrobot and Quadruped) since they require modification of SAC to incorporate multi-step returns (Barth-Maron et al., 2018), which is beyond the scope of this work. We evaluate on the remaining 15 tasks, fixing the action-repeat hyper-parameter to 2 as in Hafner et al. (2019).

We compare **DrQ** [K=2,M=2] to Dreamer (Hafner et al., 2019) and the upper-bound performance of SAC (Haarnoja et al., 2018) from states[3]. Again, we keep all the hyper-parameters of our algorithm fixed across all the tasks. In Figure 4, **DrQ** demonstrates the state-of-the-art results by collectively outperforming Dreamer (Hafner et al., 2019), although Dreamer is superior on 3 of the 15 tasks (Walker Run, Cartpole Swingup Sparse and Pendulum Swingup). On many tasks **DrQ** approaches the upper-bound performance of SAC (Haarnoja et al., 2018) trained directly on states.

---

[3]No other publicly reported results are available for the other methods due to the recency of the Dreamer (Hafner et al., 2019) benchmark.

Table 1: The PlaNet benchmark at 100k and 500k environment steps. Our method (**DrQ** [K=2,M=2]) outperforms other approaches in both the data-efficient (100k) and asymptotic performance (500k) regimes. Random shifts only version (e.g. **DrQ** [K=1,M=1]) has a competitive performance but is consistently inferior to **DrQ** [K=2,M=2], particularly for 100k steps. We emphasize, that both versions of **DrQ** use exactly the same number of interactions with both the environment and replay buffer. Note that **DrQ** [K=1,M=1] is almost identical to RAD (Laskin et al., 2020), modulo some hyper-parameter differences.

| *500k step scores* | **DrQ** [K=2,M=2] | **DrQ** [K=1,M=1] | CURL | PlaNet | SAC-AE | SLAC | SAC State |
|---|---|---|---|---|---|---|---|
| Finger Spin | **938**±103 | 913±151 | 874±151 | 718±40 | 914±107 | 771±203 | 927±43 |
| Cartpole Swingup | **868**±10 | 845±39 | 861±30 | 787±46 | 730±152 | - | 870±7 |
| Reacher Easy | **942**±71 | 857±120 | 904±94 | 588±471 | 601±135 | - | 975±5 |
| Cheetah Run | **660**±96 | 460±59 | 500±91 | 568±21 | 544±50 | 629±74 | 772±60 |
| Walker Walk | **921**±45 | 897±47 | 906±56 | 478±164 | 858±82 | 865±97 | 964±8 |
| Ball In Cup Catch | **963**±9 | 961±12 | 958±13 | 939±43 | 810±121 | 959±4 | 979±6 |
| *100k step scores* | | | | | | | |
| Finger Spin | **901**±104 | 744±144 | 779±108 | 560±77 | 747±130 | 680±130 | 672±76 |
| Cartpole Swingup | **759**±92 | 537±119 | 592±170 | 563±73 | 276±38 | - | 812±45 |
| Reacher Easy | **601**±213 | 451±210 | 517±113 | 82±174 | 225±164 | - | 919±123 |
| Cheetah Run | 344±67 | 250±58 | 307±48 | 252±173 | 240±38 | **391**±47 | 228±95 |
| Walker Walk | **612**±164 | 501±68 | 344±132 | 221±43 | 395±58 | 428±74 | 604±317 |
| Ball In Cup Catch | **913**±53 | 667±146 | 772±241 | 710±217 | 338±196 | 607±173 | 957±26 |

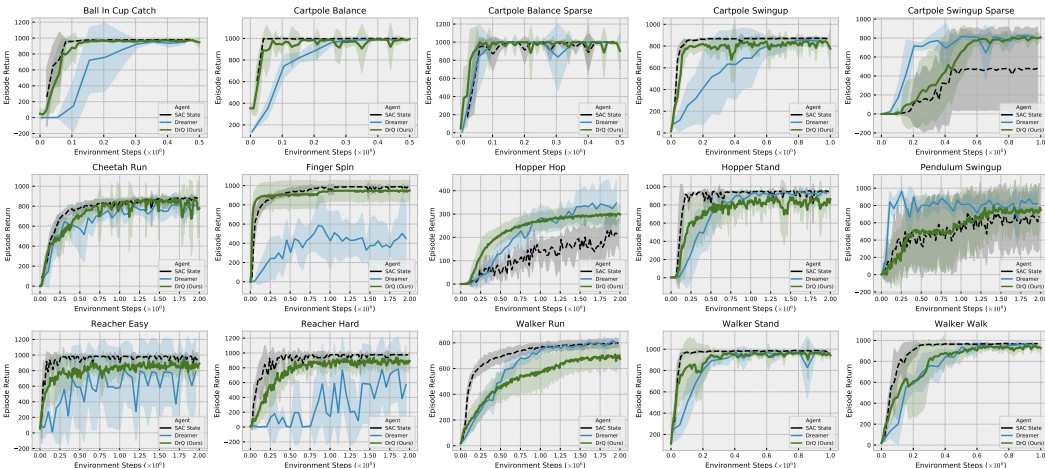

Figure 4: The Dreamer benchmark. Our method (**DrQ** [K=2,M=2]) again demonstrates superior performance over Dreamer on 12 out 15 selected tasks. In many cases it also reaches the upper-bound performance of SAC that learns directly from states.

## 5.3 ATARI 100K EXPERIMENTS

We evaluate **DrQ** [K=1,M=1] on the Atari 100k benchmark (Kaiser et al., 2019) – a sample-constrained evaluation for discrete control algorithms. The underlying RL approach to which **DrQ** is applied is a DQN, combined with double Q-learning (van Hasselt et al., 2015), n-step returns (Mnih et al., 2016), and dueling critic architecture (Wang et al., 2015). As per common practice (Kaiser et al., 2019; van Hasselt et al., 2019a), we evaluate our agent for 125k environment steps at the end of training and average its performance over 5 random seeds. Figure 5 shows the median human-normalized episode returns performance (as in Mnih et al. (2013)) of the underlying model, which we refer to as Efficient DQN, in pink. When **DrQ** is added there is a significant increase in performance (cyan), surpassing OTRainbow (Kielak, 2020) and Data Efficient Rainbow (van Hasselt et al., 2019a). **DrQ** is also superior to CURL (Srinivas et al., 2020) that uses an auxiliary loss built on top of a hybrid between OTRainbow and Efficient rainbow. **DrQ** combined with Efficient DQN thus achieves state-of-the-art performance, despite being significantly simpler than the other approaches. The experimental setup and full results are detailed in Appendix C and Appendix D respectively.

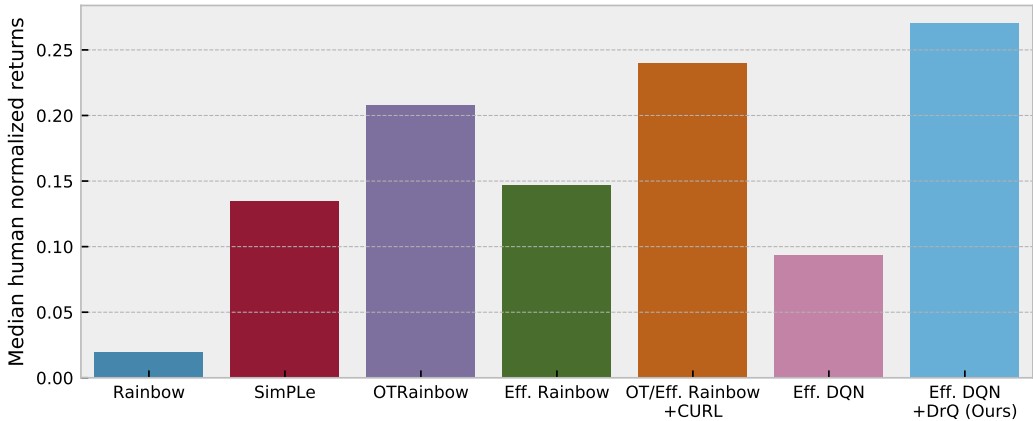

Figure 5: The Atari 100k benchmark. Compared to a set of leading baselines, our method (**DrQ** [K=1,M=1], combined with Efficient DQN) achieves the state-of-the-art performance, despite being considerably simpler. Note the large improvement that results from adding **DrQ** to Efficient DQN (pink vs cyan). By contrast, the gains from CURL, that utilizes tricks from both Data Efficient Rainbow and OTRainbow, are more modest over the underlying RL methods.

## 6 CONCLUSION

We have introduced a regularization technique, based on image shifts and Q-function augmentation, that significantly improves the performance of model-free RL algorithms trained directly from images. In contrast to the concurrent work of Laskin et al. (2020), which is a special case of **DrQ**, our method exploits the MDP structure of the problem, demonstrating gains over image augmentations alone. Our method is easy to implement and adds a negligible computational burden. We compared our method to state-of-the-art approaches on the DeepMind control suite, outperforming them on the majority of tasks and closing the gap with state-based training. On the Atari 100k benchmark **DrQ** outperforms other SOTA methods in the median metric. To the best of our knowledge, this is the first convincing demonstration of the utility of data augmentation on these standard benchmarks. Furthermore, we demonstrate the method to be robust to the choice of hyper-parameters.

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

APPENDIX

## A    EXTENDED BACKGROUND

**Reinforcement Learning from Images**    We formulate image-based control as an infinite-horizon partially observable Markov decision process (POMDP) (Bellman, 1957; Kaelbling et al., 1998). An POMDP can be described as the tuple $(\mathcal{O}, \mathcal{A}, p, r, \gamma)$, where $\mathcal{O}$ is the high-dimensional observation space (image pixels), $\mathcal{A}$ is the action space, the transition dynamics $p = Pr(o'_t|o_{\leq t}, a_t)$ capture the probability distribution over the next observation $o'_t$ given the history of previous observations $o_{\leq t}$ and current action $a_t$, $r : \mathcal{O} \times \mathcal{A} \to \mathbb{R}$ is the reward function that maps the current observation and action to a reward $r_t = r(o_{\leq t}, a_t)$, and $\gamma \in [0, 1)$ is a discount factor. Per common practice (Mnih et al., 2013), throughout the paper the POMDP is converted into an MDP (Bellman, 1957) by stacking several consecutive image observations into a state $s_t = \{o_t, o_{t-1}, o_{t-2}, \dots\}$. For simplicity we redefine the transition dynamics $p = Pr(s'_t|s_t, a_t)$ and the reward function $r_t = r(s_t, a_t)$. We then aim to find a policy $\pi(a_t|s_t)$ that maximizes the cumulative discounted return $\mathbb{E}_\pi[\sum_{t=1}^{\infty} \gamma^t r_t | a_t \sim \pi(\cdot|s_t), s'_t \sim p(\cdot|s_t, a_t), s_1 \sim p(\cdot)]$.

**Soft Actor-Critic**    The Soft Actor-Critic (SAC) (Haarnoja et al., 2018) learns a state-action value function $Q_\theta$, a stochastic policy $\pi_\theta$ and a temperature $\alpha$ to find an optimal policy for an MDP $(\mathcal{S}, \mathcal{A}, p, r, \gamma)$ by optimizing a $\gamma$-discounted maximum-entropy objective (Ziebart et al., 2008). $\theta$ is used generically to denote the parameters updated through training in each part of the model. The actor policy $\pi_\theta(a_t|s_t)$ is a parametric $\tanh$-Gaussian that given $s_t$ samples $a_t = \tanh(\mu_\theta(s_t) + \sigma_\theta(s_t)\epsilon)$, where $\epsilon \sim \mathcal{N}(0, 1)$ and $\mu_\theta$ and $\sigma_\theta$ are parametric mean and standard deviation.

The policy evaluation step learns the critic $Q_\theta(s_t, a_t)$ network by optimizing a single-step of the soft Bellman residual

$$J_Q(\mathcal{D}) = \mathbb{E}_{\substack{(s_t,a_t,s'_t)\sim\mathcal{D} \\ a'_t\sim\pi(\cdot|s'_t)}}[(Q_\theta(s_t, a_t) - y_t)^2]$$
$$y_t = r(s_t, a_t) + \gamma[Q_{\theta'}(s'_t, a'_t) - \alpha \log \pi_\theta(a'_t|s'_t)],$$

where $\mathcal{D}$ is a replay buffer of transitions, $\theta'$ is an exponential moving average of the weights as done in (Lillicrap et al., 2015). SAC uses clipped double-Q learning (van Hasselt et al., 2015; Fujimoto et al., 2018), which we omit from our notation for simplicity but employ in practice.

The policy improvement step then fits the actor policy $\pi_\theta(a_t|s_t)$ network by optimizing the objective

$$J_\pi(\mathcal{D}) = \mathbb{E}_{s_t\sim\mathcal{D}}[D_{\mathrm{KL}}(\pi_\theta(\cdot|s_t)||\exp\{\frac{1}{\alpha}Q_\theta(s_t, \cdot)\})].$$

Finally, the temperature $\alpha$ is learned with the loss

$$J_\alpha(\mathcal{D}) = \mathbb{E}_{\substack{s_t\sim\mathcal{D} \\ a_t\sim\pi_\theta(\cdot|s_t)}}[-\alpha \log \pi_\theta(a_t|s_t) - \alpha\bar{\mathcal{H}}],$$

where $\bar{\mathcal{H}} \in \mathbb{R}$ is the target entropy hyper-parameter that the policy tries to match, which in practice is usually set to $\bar{\mathcal{H}} = -|\mathcal{A}|$.

**Deep Q-learning**    DQN (Mnih et al., 2013) also learns a convolutional neural net to approximate Q-function over states and actions. The main difference is that DQN operates on discrete actions spaces, thus the policy can be directly inferred from Q-values. The parameters of DQN are updated by optimizing the squared residual error

$$J_Q(\mathcal{D}) = \mathbb{E}_{(s_t,a_t,s'_t)\sim\mathcal{D}}[(Q_\theta(s_t, a_t) - y_t)^2]$$
$$y_t = r(s_t, a_t) + \gamma \max_{a'} Q_{\theta'}(s'_t, a').$$

In practice, the standard version of DQN is frequently combined with a set of tricks that improve performance and training stability, wildly known as Rainbow (van Hasselt et al., 2015).

## B  THE DEEPMIND CONTROL SUITE EXPERIMENTS SETUP

Our PyTorch SAC (Haarnoja et al., 2018) implementation is based off of Yarats & Kostrikov (2020).

### B.1  ACTOR AND CRITIC NETWORKS

We employ clipped double Q-learning (van Hasselt et al., 2015; Fujimoto et al., 2018) for the critic, where each $Q$-function is parametrized as a 3-layer MLP with ReLU activations after each layer except of the last. The actor is also a 3-layer MLP with ReLUs that outputs mean and covariance for the diagonal Gaussian that represents the policy. The hidden dimension is set to 1024 for both the critic and actor.

### B.2  ENCODER NETWORK

We employ an encoder architecture from Yarats et al. (2019). This encoder consists of four convolutional layers with $3 \times 3$ kernels and 32 channels. The ReLU activation is applied after each conv layer. We use stride to 1 everywhere, except of the first conv layer, which has stride 2. The output of the convnet is feed into a single fully-connected layer normalized by LayerNorm (Ba et al., 2016). Finally, we apply tanh nonlinearity to the 50 dimensional output of the fully-connected layer. We initialize the weight matrix of fully-connected and convolutional layers with the orthogonal initialization (Saxe et al., 2013) and set the bias to be zero.

The actor and critic networks both have separate encoders, although we share the weights of the conv layers between them. Furthermore, only the critic optimizer is allowed to update these weights (e.g. we stop the gradients from the actor before they propagate to the shared conv layers).

### B.3  TRAINING AND EVALUATION SETUP

Our agent first collects 1000 seed observations using a random policy. The further training observations are collected by sampling actions from the current policy. We perform one training update every time we receive a new observation. In cases where we use action repeat, the number of training observations is only a fraction of the environment steps (e.g. a 1000 steps episode at action repeat 4 will only results into 250 training observations). We evaluate our agent every 10000 true environment steps by computing the average episode return over 10 evaluation episodes. During evaluation we take the mean policy action instead of sampling.

### B.4  PLANET AND DREAMER BENCHMARKS

We consider two evaluation setups that were introduced in PlaNet (Hafner et al., 2018) and Dreamer (Hafner et al., 2019), both using tasks from the DeepMind control suite (Tassa et al., 2018). The PlaNet benchmark consists of six tasks of various traits. Importantly, the benchmark proposed to use a different action repeat hyper-parameter for each task, which we summarize in Table 2.

The Dreamer benchmark considers an extended set of tasks, which makes it more difficult that the PlaNet setup. Additionally, this benchmark requires to use the same set hyper-parameters for each task, including action repeat (set to 2), which further increases the difficulty.

Table 2: The action repeat hyper-parameter used for each task in the PlaNet benchmark.

| Task name | Action repeat |
|---|---|
| Cartpole Swingup | 8 |
| Reacher Easy | 4 |
| Cheetah Run | 4 |
| Finger Spin | 2 |
| Ball In Cup Catch | 4 |
| Walker Walk | 2 |

### B.5 PIXELS PREPROCESSING

We construct an observational input as an 3-stack of consecutive frames (Mnih et al., 2013), where each frame is a RGB rendering of size $84 \times 84$ from the 0th camera. We then divide each pixel by 255 to scale it down to $[0, 1]$ range.

### B.6 OTHER HYPER PARAMETERS

Due to computational constraints for all the continuous control ablation experiments in the main paper and appendix we use a minibatch size of 128, while for the main results we use minibatch of size 512. In Table 3 we provide a comprehensive overview of all the other hyper-parameters.

Table 3: An overview of used hyper-parameters in the DeepMind control suite experiments.

| Parameter | Setting |
|---|---|
| Replay buffer capacity | 100000 |
| Seed steps | 1000 |
| Ablations minibatch size | 128 |
| Main results minibatch size | 512 |
| Discount $\gamma$ | 0.99 |
| Optimizer | Adam |
| Learning rate | $10^{-3}$ |
| Critic target update frequency | 2 |
| Critic Q-function soft-update rate $\tau$ | 0.01 |
| Actor update frequency | 2 |
| Actor log stddev bounds | $[-10, 2]$ |
| Init temperature | 0.1 |

# C   THE ATARI 100K EXPERIMENTS SETUP

For ease of reproducibility in Table 4 we report the hyper-parameter settings used in the Atari 100k experiments. We largely reuse the hyper-parameters from OTRainbow (Kielak, 2020), but adapt them for DQN (Mnih et al., 2013). Per common practise, we average performance of our agent over 5 random seeds. The evaluation is done for 125k environment steps at the end of training for 100k environment steps.

Table 4: A complete overview of hyper parameters used in the Atari 100k experiments.

| Parameter | Setting |
|---|---|
| Data augmentation | Random shifts and Intensity |
| Grey-scaling | True |
| Observation down-sampling | $84 \times 84$ |
| Frames stacked | 4 |
| Action repetitions | 4 |
| Reward clipping | $[-1, 1]$ |
| Terminal on loss of life | True |
| Max frames per episode | 108k |
| Update | Double Q |
| Dueling | True |
| Target network: update period | 1 |
| Discount factor | 0.99 |
| Minibatch size | 32 |
| Optimizer | Adam |
| Optimizer: learning rate | 0.0001 |
| Optimizer: $\beta_1$ | 0.9 |
| Optimizer: $\beta_2$ | 0.999 |
| Optimizer: $\epsilon$ | 0.00015 |
| Max gradient norm | 10 |
| Training steps | 100k |
| Evaluation steps | 125k |
| Min replay size for sampling | 1600 |
| Memory size | Unbounded |
| Replay period every | 1 step |
| Multi-step return length | 10 |
| Q network: channels | $32, 64, 64$ |
| Q network: filter size | $8 \times 8, 4 \times 4, 3 \times 3$ |
| Q network: stride | $4, 2, 1$ |
| Q network: hidden units | 512 |
| Non-linearity | ReLU |
| Exploration | $\epsilon$-greedy |
| $\epsilon$-decay | 5000 |

## D  FULL ATARI 100K RESULTS

Besides reporting in Figure 5 median human-normalized episode returns over the 26 Atari games used in (Kaiser et al., 2019), we also provide the mean episode return for each individual game in Table 5.

Table 5: Mean episode returns on each of 26 Atari games from the setup in Kaiser et al. (2019). The results are recorded at the end of training and averaged across 5 random seeds (the CURL's results are averaged over 3 seeds as reported in Srinivas et al. (2020)). On each game we mark as bold the highest score. Our method demonstrates better overall performance (as reported in Figure 5).

| Game | Rainbow | SimPLe | OTRainbow | Eff. Rainbow | OT/Eff. Rainbow +CURL | Eff. DQN | Eff. DQN +DrQ (Ours) |
|---|---|---|---|---|---|---|---|
| Alien | 318.7 | 616.9 | 824.7 | 739.9 | **1148.2** | 558.1 | 702.5 |
| Amidar | 32.5 | 88.0 | 82.8 | 188.6 | **232.3** | 63.7 | 100.2 |
| Assault | 231.0 | 527.2 | 351.9 | 431.2 | 543.7 | **589.5** | 490.3 |
| Asterix | 243.6 | **1128.3** | 628.5 | 470.8 | 524.3 | 341.9 | 577.9 |
| BankHeist | 15.6 | 34.2 | 182.1 | 51.0 | 193.7 | 74.0 | **205.3** |
| BattleZone | 2360.0 | 5184.4 | 4060.6 | 10124.6 | **11208.0** | 4760.8 | 6240.0 |
| Boxing | -24.8 | **9.1** | 2.5 | 0.2 | 4.8 | -1.8 | 5.1 |
| Breakout | 1.2 | 16.4 | 9.8 | 1.9 | **18.2** | 7.3 | 14.3 |
| ChopperCommand | 120.0 | **1246.9** | 1033.3 | 861.8 | 1198.0 | 624.4 | 870.1 |
| CrazyClimber | 2254.5 | **62583.6** | 21327.8 | 16185.3 | 27805.6 | 5430.6 | 20072.2 |
| DemonAttack | 163.6 | 208.1 | 711.8 | 508.0 | 834.0 | 403.5 | **1086.0** |
| Freeway | 0.0 | 20.3 | 25.0 | **27.9** | **27.9** | 3.7 | 20.0 |
| Frostbite | 60.2 | 254.7 | 231.6 | 866.8 | **924.0** | 202.9 | 889.9 |
| Gopher | 431.2 | 771.0 | 778.0 | 349.5 | **801.4** | 320.8 | 678.0 |
| Hero | 487.0 | 2656.6 | 6458.8 | **6857.0** | 6235.1 | 2200.1 | 4083.7 |
| Jamesbond | 47.4 | 125.3 | 112.3 | 301.6 | **400.1** | 133.2 | 330.3 |
| Kangaroo | 0.0 | 323.1 | 605.4 | 779.3 | 345.3 | 448.6 | **1282.6** |
| Krull | 1468.0 | **4539.9** | 3277.9 | 2851.5 | 3833.6 | 2999.0 | 4163.0 |
| KungFuMaster | 0.0 | **17257.2** | 5722.2 | 14346.1 | 14280.0 | 2020.9 | 7649.0 |
| MsPacman | 67.0 | 1480.0 | 941.9 | 1204.1 | **1492.8** | 872.0 | 1015.9 |
| Pong | -20.6 | **12.8** | 1.3 | -19.3 | 2.1 | -19.4 | -17.1 |
| PrivateEye | 0.0 | 58.3 | 100.0 | 97.8 | 105.2 | **351.3** | -50.4 |
| Qbert | 123.5 | **1288.8** | 509.3 | 1152.9 | 1225.6 | 627.5 | 769.1 |
| RoadRunner | 1588.5 | 5640.6 | 2696.7 | **9600.0** | 6786.7 | 1491.9 | 8296.3 |
| Seaquest | 131.7 | **683.3** | 286.9 | 354.1 | 408.0 | 240.1 | 299.4 |
| UpNDown | 504.6 | **3350.3** | 2847.6 | 2877.4 | 2735.2 | 2901.7 | 3134.8 |
| Median human-normalised episode returns | 0.020 | 0.135 | 0.208 | 0.147 | 0.240 | 0.094 | **0.270** |

## E  IMAGE AUGMENTATIONS ABLATION

Following (Chen et al., 2020), we evaluate popular image augmentation techniques, namely random shifts, cutouts, vertical and horizontal flips, random rotations and imagewise intensity jittering. Below, we provide a comprehensive overview of each augmentation. Furthermore, we examine effectiveness of these techniques in Figure 6.

**Random Shift**  We bring our attention to random shifts that are commonly used to regularize neural networks trained on small images (Becker & Hinton, 1992; Simard et al., 2003; LeCun et al., 1989; Ciresan et al., 2011; Ciregan et al., 2012). In our implementation of this method images of size $84 \times 84$ are padded each side by 4 pixels (by repeating boundary pixels) and then randomly cropped back to the original $84 \times 84$ size.

**Cutout**  Cutouts introduced in DeVries & Taylor (2017) represent a generalization of Dropout (Hinton et al., 2012). Instead of masking individual pixels cutouts mask square regions. Since image pixels can be highly correlated, this technique is proven to improve training of neural networks.

**Horizontal/Vertical Flip**  This technique simply flips an image either horizontally or vertically with probability $0.1$.

**Rotate**  Here, an image is rotated by $r$ degrees, where $r$ is uniformly sampled from $[-5, -5]$.

**Intensity** Each $N \times C \times 84 \times 84$ image tensor is multiplied by a single scalar $s$, which is computed as $s = \mu + \sigma \cdot \mathrm{clip}(r, -2, 2)$, where $r \sim \mathcal{N}(0, 1)$. For our experiments we use $\mu = 1.0$ and $\sigma = 0.1$.

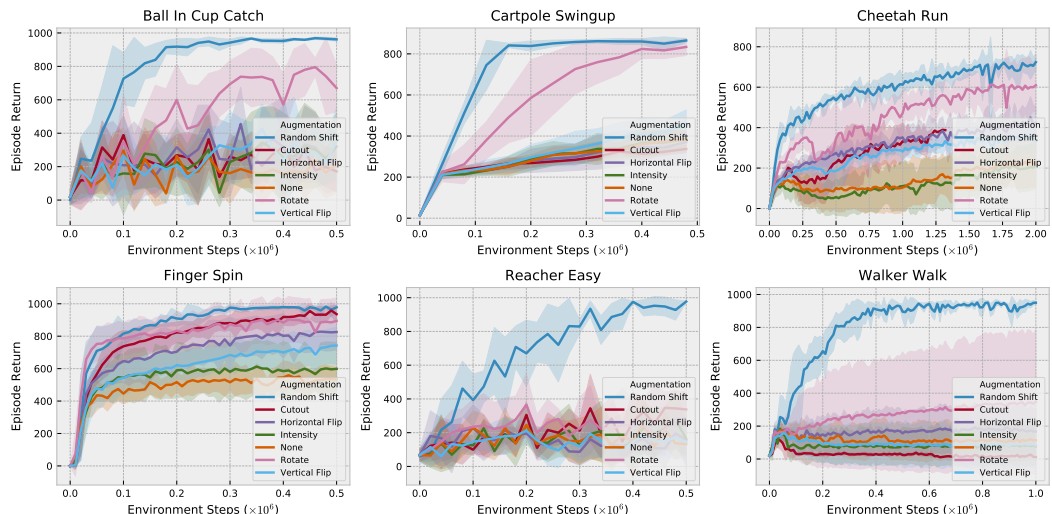

Figure 6: Various image augmentations have different effect on the agent's performance. Overall, we conclude that using image augmentations helps to fight overfitting. Moreover, we notice that random shifts proven to be the most effective technique for tasks from the DeepMind control suite.

**Implementation** Finally, we provide Python-like implementation for the aforementioned augmentations powered by Kornia (Riba et al., 2020).

```python
import torch
import torch.nn as nn
import kornia.augmentation as aug

random_shift = nn.Sequential(nn.ReplicationPad2d(4),aug.RandomCrop((84, 84)))

cutout = aug.RandomErasing(p=0.5)

h_flip = aug.RandomHorizontalFlip(p=0.1)

v_flip = aug.RandomVerticalFlip(p=0.1)

rotate = aug.RandomRotation(degrees=5.0)

intensity = Intensity(scale=0.1)

class Intensity(nn.Module):
  def __init__(self, scale):
    super().__init__()
    self.scale = scale

  def forward(self, x):
    r = torch.randn((x.size(0), 1, 1, 1), device=x.device)
    noise = 1.0 + (self.scale * r.clamp(-2.0, 2.0))
    return x * noise
```

## F   K AND M HYPER-PARAMETERS ABLATION

We further ablate the K,M hyper-parameters from Algorithm 1 to understand their effect on performance. In Figure 7 we observe that increase values of K,M improves the agent's performance. We choose to use the [K=2,M=2] parametrization as it strikes a good balance between performance and computational demands.

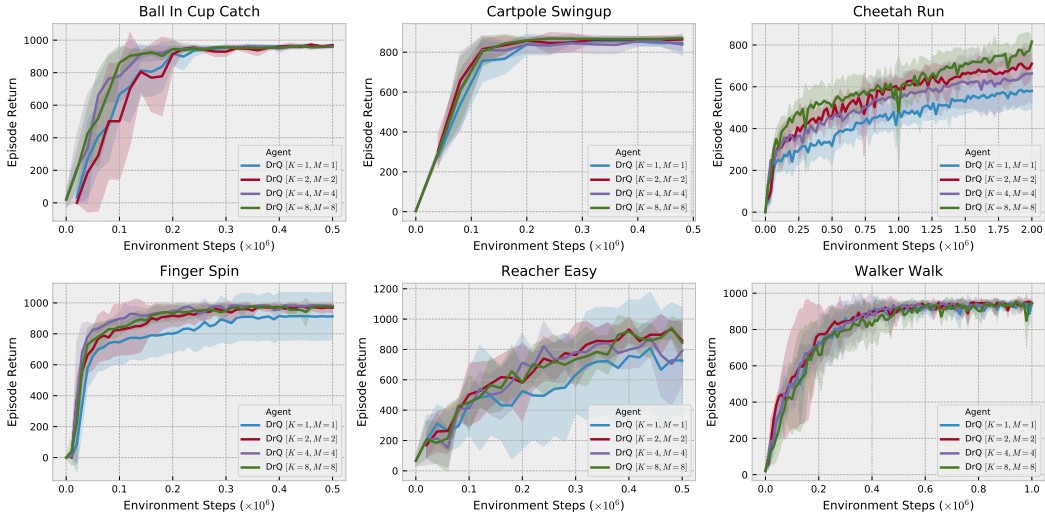

Figure 7: Increasing values of K,M hyper-parameters generally correlates positively with the agent's performance, especially on the harder tasks, such as Cheetah Run.

## G   ROBUSTNESS INVESTIGATION

To demonstrate the robustness of our approach (Henderson et al., 2018), we perform a comprehensive study on the effect different hyper-parameter choices have on performance. A review of prior work (Hafner et al., 2018; 2019; Lee et al., 2019; Srinivas et al., 2020) shows consistent values for discount $\gamma = 0.99$ and target update rate $\tau = 0.01$ parameters, but variability on network architectures, mini-batch sizes, learning rates. Since our method is based on SAC (Haarnoja et al., 2018), we also check whether the initial value of the temperature is important, as it plays a crucial role in the initial phase of exploration. We omit search over network architectures since Figure 1b shows our method to be robust to the exact choice. We thus focus on three hyper-parameters: mini-batch size, learning rate, and initial temperature.

Due to computational demands, experiments are restricted to a subset of tasks from Tassa et al. (2018): Walker Walk, Cartpole Swingup, and Finger Spin. These were selected to be diverse, requiring different behaviors including locomotion and goal reaching. A grid search is performed over mini-batch sizes $\{128, 256, 512\}$, learning rates $\{0.0001, 0.0005, 0.001, 0.005\}$, and initial temperatures $\{0.005, 0.01, 0.05, 0.1\}$. We follow the experimental setup from Appendix B, except that only 3 seeds are used due to the computation limitations, but since variance is low the results are representative.

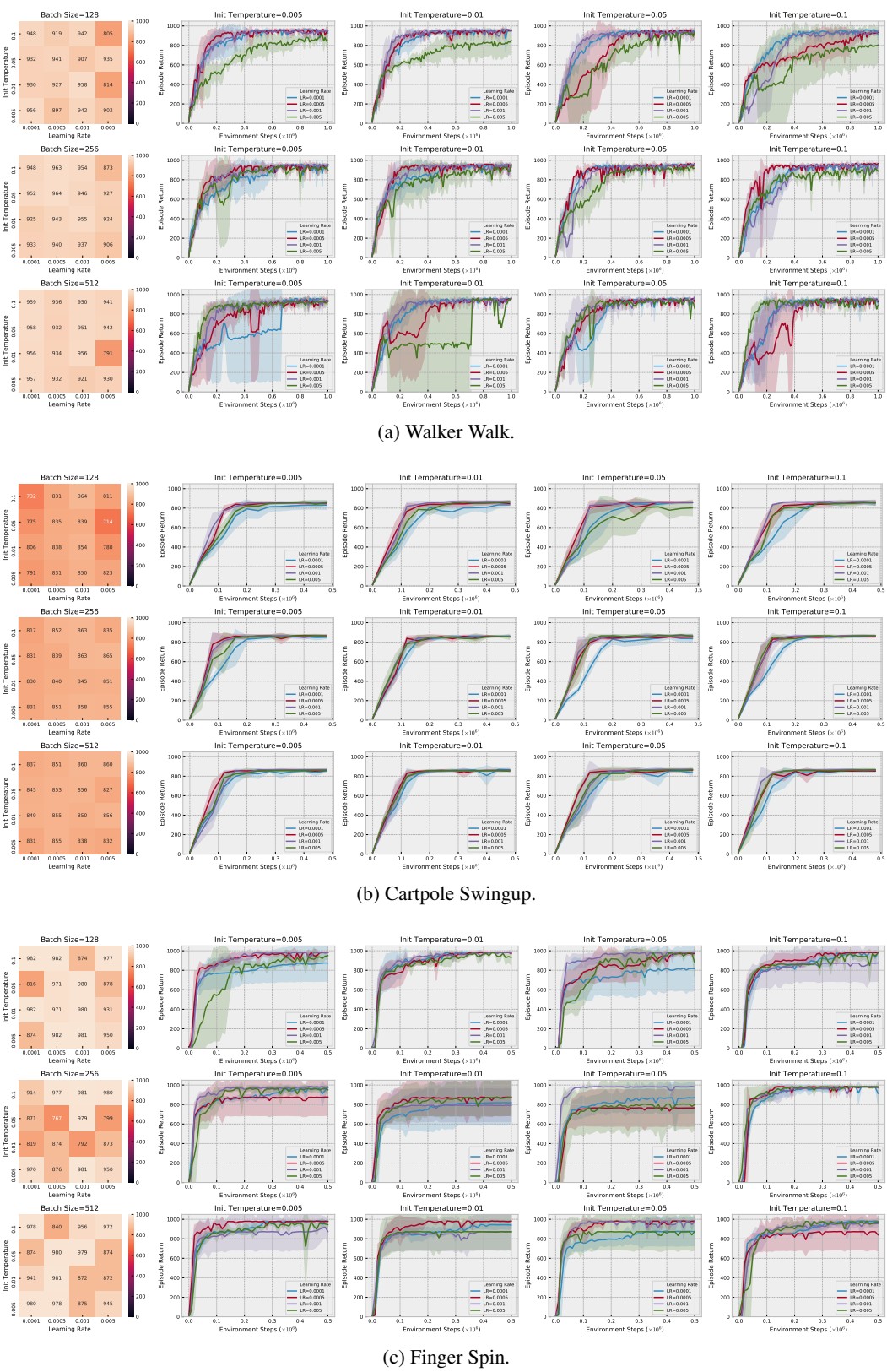

Figure 8: A robustness study of our algorithm (**DrQ**) to changes in mini-batch size, learning rate, and initial temperature hyper-parameters on three different tasks from (Tassa et al., 2018). Each row corresponds to a different mini-batch size. The low variance of the curves and heat-maps shows **DrQ** to be generally robust to exact hyper-parameter settings.

Figure 8 shows performance curves for each configuration as well as a heat map over the mean performance of the final evaluation episodes, similar to Mnih et al. (2016). Our method demonstrates good stability and is largely invariant to the studied hyper-parameters. We emphasize that for simplicity the experiments in Section 5 use the default learning rate of Adam (Kingma & Ba, 2014) (0.001), even though it is not always optimal.

# H  IMPROVED DATA-EFFICIENT REINFORCEMENT LEARNING FROM PIXELS

Our method allows to generate many various transformations from a training observation due to the data augmentation strategy. Thus, we further investigate whether performing more training updates per an environment step can lead to even better sample-efficiency. Following van Hasselt et al. (2019b) we compare a single update with a mini-batch of 512 transitions with 4 updates with 4 different mini-batches of size 128 samples each. Performing more updates per an environment step leads to even worse over-fitting on some tasks without data augmentation (see Figure 9a), while our method **DrQ**, that takes advantage of data augmentation, demonstrates improved sample-efficiency (see Figure 9b).

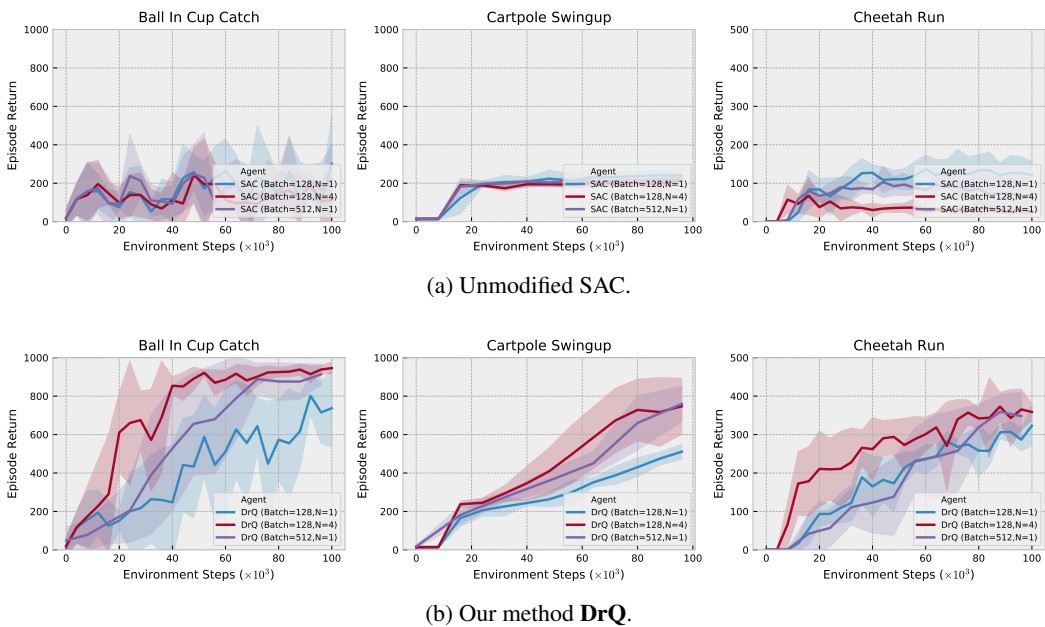

(a) Unmodified SAC.

(b) Our method **DrQ**.

Figure 9: In the data-efficient regime, where we measure performance at 100k environment steps, **DrQ** is able to enhance its efficiency by performing more training iterations per an environment step. This is because **DrQ** allows to generate various transformations for a training observation.

