# OpenReview forum: "Image Augmentation Is All You Need: Regularizing Deep Reinforcement Learning from Pixels"
_ICLR.cc/2021/Conference — ICLR 2021 Spotlight_

### Official Review · AnonReviewer4 · 2020-10-28
**The paper proposes a new approach in RL by showing effectiveness of image augmentation in DQN. They compares against benchmark available in DeepMind control suite on several environments.**

**Rating:** 7
**Confidence:** 4

**Review:**

PROS
-------------------------------------------
- Finding that image augmentation helps in learning a good policy is indeed a substantial contributions.
- Other two aspects to regularize Q-values are also helping to learn a better policy.

CONS
--------------------------------------------
The paper presents many experiments but there are a few crucial ones which are missing.
For example,
- what is the impact on training time of DrQ as compared conventional DQN?
- How much augmentation is good? etc.

Question that needs justification:
---------------------------------------------
- Overall, the paper is well written and explain various things but the algo-1 needs explanation of all parameters. For example, \math{D} (reader has to think that it's a replay buffer), \math{u(T)} is not at all clear. Many notation are not explained even in text and needs a clear explanation for reader from RL and non-RL domain.

- The primary claim of the paper is that image augmentation improves the performance. Sec5.1 shows significant improvement when image augmentation is used with different methods but it is very strange to see the improvement is just by adding 4 pixels on image boundary. Is there implication when we do more augmentation by increasing the size of random shift? How does the augmented image compare visually to the original image? The figure-6 in appendix shows all the results but which one is the random shift?

- In figure4, the SAC state is significantly better than the DrQ and no explanation is provided. Why one should not used SAC compared to DrQ?

- Paper claims that the proposed method can work with any model-free RL algorithm. Any justification or experiments to support the claim? If not, the contribution needs a re-writing.

---

> ### Author Response · Authors · 2020-11-10
> **Review response -- thanks for the feedback!**
>
> We thank the reviewer for their encouraging and constructive feedback and deeming our work as a “significant contribution”.
>
> Below we would like to address the reviewer’s concerns about work:
>
>
> #1
> Q: The paper presents many experiments but there are a few crucial ones which are missing. For example, what is the impact on training time of DrQ as compared conventional DQN?
>
> A: The impact on training times depends on the choice of the K and M hyperparameters. For example, DrQ[K=1,M=1] incurs only negligible computational cost that is proportional to image augmentation itself and minor compared to the cost of the entire training. DrQ[K=2,M=2] is twice as slow as DrQ[K=1,M=1], but it is still 5-10x faster than Dreamer or PlaNet (because they learn a multistep model), and about the same as CURL and SAC-AE (because they have an additional auxiliary loss).
>
> #2
> Q: How much augmentation is good? Etc.
>
> A: In RL one has to be careful with data augmentation, and only use those that don’t change the underlying MDP of the task (e.g. reward function, observational function, or transition probability), there are no such limitations in CV. To this end, we formulate the notion of optimality invariant image transformations in Section 3.2, which allows us to qualify only a few data augmentations from the set of canonical augmentations in CV.
>
> #3
> Q: Overall, the paper is well written and explain various things but the algo-1 needs explanation of all parameters. For example, \math{D} (reader has to think that it's a replay buffer), \math{u(T)} is not at all clear. Many notation are not explained even in text and needs a clear explanation for reader from RL and non-RL domain.
>
> A: Thank you, we will add additional clarification.
>
> #4
> Q: The primary claim of the paper is that image augmentation improves the performance. Sec5.1 shows significant improvement when image augmentation is used with different methods but it is very strange to see the improvement is just by adding 4 pixels on image boundary. Is there implication when we do more augmentation by increasing the size of random shift? How does the augmented image compare visually to the original image? The figure-6 in appendix shows all the results but which one is the random shift?
>
> A: 4 pixels are added to the image boundary only to perform a valid random cut. Empirically, we found that 4 performs the best compared to larger shifts (those lead to parts of the anget disappearing from an observation) and smaller shifts (providing very little randomization). A randomly shifted by 4 pixels observation looks pretty much identical to the original observation, the only difference is that the agent is shifted a bit into a random direction. In Figure 6 the blue line (Random Shift) corresponds to performing the 4 pixels random shift augmentation, which serves as our primary image augmentation throughout the paper.
>
> #5
> Q: In figure4, the SAC state is significantly better than the DrQ and no explanation is provided. Why one should not used SAC compared to DrQ?
>
> A: DrQ is concerned with learning to control from *image pixels* as input. SAC State has access to privileged information, such as internal states of the system, which perfectly explains the underlying MDP. As such it represents an upper-bound on performance achievable when learning from pixels. In virtually all practical settings (e.g. robotics), internal state information is not available and a learning algorithm can only rely on high-dimensional image observations of the environment, which makes learning considerably harder. Note that vanilla SAC from pixels fails in many cases (see Figure 1). We will further clarify this in the camera ready.
>
> #6
> Q: Paper claims that the proposed method can work with any model-free RL algorithm. Any justification or experiments to support the claim? If not, the contribution needs a re-writing.
>
> A: After the first draft of our work was made available on arxiv and before the current version was submitted to ICLR, we noticed many researchers from various groups adopted our data augmentation techniques in their work. Specifically, inspired by our work our data augmentation technique was combined with PPO (https://arxiv.org/pdf/2006.12862.pdf) and Dreamer (https://arxiv.org/pdf/2007.14535.pdf). For these reasons, we decided to omit this experiment in our submission as follow up literature comprehensively studies this question. To summarize, we now have strong empirical evidence that data augmentation is beneficial in image-based RL across four of the most widely adopted RL algorithms (SAC, DQN, PPO, Dreamer). We will reference and discuss these methods in the next revision of our work.
>
>
> In the light of these clarifications we would appreciate it if the reviewer confirmed that all their concerns had been addressed and, if so, reconsider their assessment.

---

### Official Review · AnonReviewer3 · 2020-10-28
**interesting paper on tackling RL with data augmentation**

**Rating:** 7
**Confidence:** 3

**Review:**

This paper tackle the effectiveness of data augmentation in reinforcement learning. Authors have introduced a regularization technique, based on image shifts and Q-function augmentation,(DrQ) that significantly improves the performance of model-free RL algorithms trained directly from images.

Here are two positive points in the paper:
1. The authors have combined a model-free RL and regularized method, which shows a promising direction with a training strategy with augmented data.
2. The authors have solid improve in continuous control by image pixel.

The negative point is that their DrQ method is very rough and very intuitive, authors can tackle more on the transformations of the images.

Generally the paper is acceptable and they have shown a promising direction.

---

> ### Author Response · Authors · 2020-11-10
> **Review response -- thanks for the feedback!**
>
> We thank the reviewer for their encouraging and constructive feedback and acknowledging significant performance improvement demonstrated by our method on image-based RL tasks.
>
> Below we would like to address the reviewer’s concerns about work:
>
>
> Q: The negative point is that their DrQ method is very rough and very intuitive, authors can tackle more on the transformations of the images.
>
> A: We respectfully disagree with the characterization that our method is “rough”. On the contrary, we strongly believe that the simplicity and intuitiveness of our method are its main strengths that will facilitate its wide adoption. Furthermore, we would like to point out that we have provided a range of ablations that demonstrate the principled manner in which we arrived at the final algorithm. Theoretical analysis of data augmentation is a clear direction for future work, which we note to be an open and challenging problem (e.g. see https://arxiv.org/abs/1907.10905).
>
>
> We would appreciate it if the reviewer would be willing to reconsider their assessment and/or provide us with further feedback.

---

### Official Review · AnonReviewer1 · 2020-10-29
**Official Blind Review #1**

**Rating:** 7
**Confidence:** 5

**Review:**

 ##########################################################################

Summary:

This paper investigates data augmentation in the context of RL and proposes a novel augmentation algorithm to enabling robust learning directly from pixels without the need for auxiliary losses or pre-training. The authors propose to average both the Q function and its target over multiple image transformations. The experiments on DeepMind control suite and Atari 100k benchmark show that their method outperforms previous model-free, model-based and contrastive learning approaches.

##########################################################################

Pros:

1. This paper tackles a valuable problem of improving RL by data augmentation. It will have a broad impact on the area of both representation learning and reinforcement learning.

2. The idea of averaging both the Q function and its target over multiple image transformations is interesting and promising. This approach is easy to use and can be combined with any model-free RL algorithm.

3. The paper is well written and the results section is well structured. They outperform baseline methods on two popular benchmarks and conduct ablation studies to verify the contribution of each component.

##########################################################################

Cons:

1. The proposed idea is very similar to RAD, a concurrent work by Laskin et al. The performance is also similar between these two approaches. More discussions and comparisons will help the readers better understand the difference.

2. They claim data augmentation is all you need. To support this strong claim. I think the authors should conduct more experiments on more base algorithms. Can we get the same conclusion if we add the data augmentation techniques to other model-free RL algorithms (e.g., PPO or TD3) or a model-based RL algorithm (e.g., PlaNet or Dreamer)?

##########################################################################

Taken both pros and cons in to consideration, I vote for an acceptance because of the novelty of the proposed idea and large-scale comparisons to previous model-free, model-based and contrastive approaches. However, the experiments in the paper are not sufficient to support their claims “data augmentation is all you need” and I do suggest the authors to include more experiments to make it clear.

---

> ### Author Response · Authors · 2020-11-10
> **Review response -- thanks for the feedback!**
>
> We thank the reviewer for their encouraging and constructive feedback. We especially appreciate that the reviewer thinks that our work “will have a broad impact on the area of both representation learning and reinforcement learning”, in fact, we are already seeing significant interest and adoption of our method in the community (30+ citations according to Semantic Scholar in less than 6 months).
>
> Below we would like to address the reviewer’s concerns about work:
>
> #1
> Q:“The proposed idea is very similar to RAD, a concurrent work by Laskin et al. The performance is also similar between these two approaches. More discussions and comparisons will help the readers better understand the difference.”
>
> A: We would like to reiterate that RAD is a strict subset of our method and can be instantiated by using DrQ[K=1,M=1]. Furthermore, our manuscript was made publicly available on arxiv prior to RAD, which was developed concurrently and independently from us. We will add additional discussion that clarifies this point.
>
>
> #2 Q: They claim data augmentation is all you need. To support this strong claim. I think the authors should conduct more experiments on more base algorithms. Can we get the same conclusion if we add the data augmentation techniques to other model-free RL algorithms (e.g., PPO or TD3) or a model-based RL algorithm (e.g., PlaNet or Dreamer)?
>
> A: After the first draft of our work was made available on arxiv and before the current version was submitted to ICLR, we noticed many researchers from various groups adopted our data augmentation techniques in their work. Specifically, inspired by our work our data augmentation technique was combined with PPO (https://arxiv.org/pdf/2006.12862.pdf) and Dreamer (https://arxiv.org/pdf/2007.14535.pdf). For these reasons, we decided to omit this experiment in our submission as follow up literature comprehensively studies this question. To summarize, we now have strong empirical evidence that data augmentation is beneficial in image-based RL across four of the most widely adopted RL algorithms (SAC, DQN, PPO, Dreamer). We will reference and discuss these methods in the next revision of our work.
>
>
>
> In the light of these clarifications we would appreciate it if the reviewer confirmed that all their concerns had been addressed and, if so, reconsider their assessment.

---

> > ### Comment · AnonReviewer1 · 2020-11-11
> > **Thanks for the feedback.**
> >
> > Thanks for the feedback. My concerns are addressed and I will raise my score to an "accept". Looking forward to your next version!

---

### Official Review · AnonReviewer2 · 2020-11-01

**Rating:** 7
**Confidence:** 5

**Review:**

Summary: To enable robust policy learning with image observations, the paper proposes a simple data augmentation technique that can be used with existing model-free reinforcement learning algorithms. It defines a notion of optimality invariant state transformation which preserves the Q function. An example of such transformations can be random image translations. It uses these transformations to (i) transform the input images, (ii) average the target Q values, and (iii) average the Q function themselves. Using this simple technique, they are able to get SOTA on DM control tasks and Atari 100k benchmark. On DM control tasks, it’s able to outperform SAC trained on state representations. Additionally, the paper provides ablation studies on different image transformations and robustness analysis with respect to hyperparameter settings.

Novelty: While data augmentation techniques are common in computer vision, this was the first work (concurrently with RAD) to apply the technique in the context of reinforcement learning.

Reasons for score: Overall, I vote for accepting. The reasons are as follows. (i) It’s a simple technique which can be used with any RL algorithm to improve the performance of the algorithm (ii) Good and detailed evaluation  (iii) additional ablation studies and robustness analysis present

Questions: Will using different random image transformations in sequence help? (My hunch is it won’t as shown in RAD but still wanted your view given this method also does the average of Q function and target Q values)

---

> ### Author Response · Authors · 2020-11-10
> **Review response -- thanks for the feedback!**
>
> We thank the reviewer for their encouraging and constructive feedback. We especially appreciate that the reviewer acknowledged the novelty and simplicity of our method.
>
> Below we would like to address the reviewer’s concerns about work:
>
>
> Q: Will using different random image transformations in sequence help? (My hunch is it won’t as shown in RAD but still wanted your view given this method also does the average of Q function and target Q values)
>
> A: We concur with the reviewer that this is an interesting idea and something that we also tried. Unfortunately, our empirical study of using different transformations in sequence has only demonstrated insignificant difference in performance, thus we omitted these results for conciseness. We will add these results to the appendix in the camera ready.
>
>
> We would appreciate it if the reviewer would be willing to reconsider their assessment and/or provide us with further feedback.

---

### Author Response · Authors · 2020-11-24
**Paper Update**

We would like to thank the reviewers again for the constructive feedback which has helped us to further improve our paper. We have uploaded a revised draft which contains the following updates:

* Added additional references and discussion to some follow up works that use our data augmentation.
* Added more detailed comments to the algorithm.

We hope these changes, together with our responses below, fully address the reviewers' questions.

---

### Decision · Program_Chairs · 2021-01-07
**Final Decision**

**Decision:**

Accept (Spotlight)

**Comment:**

The paper describes a new data augmentation approach for image based RL.  The approach is both simple and effective.  It improves significantly the performance of several algorithms across a number of tasks.  The reviewers were unanimous about the benefits of the proposed technique.  This represents an important advance for RL.